# Multivariate genomic scan implicates novel loci and haem metabolism in human ageing

Paul R. H. J. Timmers [1]✉, James F. Wilson [1,2], Peter K. Joshi [1,5]✉ & Joris Deelen [3,4,5]✉

Ageing phenotypes, such as years lived in good health (healthspan), total years lived (lifespan), and survival until an exceptional old age (longevity), are of interest to us all but require exceptionally large sample sizes to study genetically. Here we combine existing genome-wide association summary statistics for healthspan, parental lifespan, and longevity in a multivariate framework, increasing statistical power, and identify 10 genomic loci which influence all three phenotypes, of which five (near *FOXO3*, *SLC4A7*, *LINC02513*, *ZW10*, and *FGD6*) have not been reported previously at genome-wide significance. The majority of these 10 loci are associated with cardiovascular disease and some affect the expression of genes known to change their activity with age. In total, we implicate 78 genes, and find these to be enriched for ageing pathways previously highlighted in model organisms, such as the response to DNA damage, apoptosis, and homeostasis. Finally, we identify a pathway worthy of further study: haem metabolism.

[1] Centre for Global Health Research, Usher Institute, University of Edinburgh, Edinburgh, UK. [2] MRC Human Genetics Unit, Institute of Genetics and Molecular Medicine, University of Edinburgh, Edinburgh, UK. [3] Max Planck Institute for Biology of Ageing, Cologne, Germany. [4] Molecular Epidemiology, Department of Biomedical Data Sciences, Leiden University Medical Center, Leiden, The Netherlands. [5] These authors contributed equally: Peter K. Joshi, Joris Deelen. ✉email: paul.timmers@ed.ac.uk; peter.joshi@ed.ac.uk; joris.deelen@age.mpg.de

Human ageing is characterised by a progressive decline in the ability to maintain homeostasis, leading to the onset of age-related diseases and eventually death. However, there is much variation between individuals, with some experiencing chronic disease early on and dying before age 60, while others being able to reach an exceptional old age, often free of disease until the last few years of life[1]. A long and healthy life is determined by many different factors, including lifestyle, environment, genetics, and pure chance. Recent estimates suggest the genetic components of both human lifespan (i.e. the number of years lived) and healthspan (the number of years lived in good health free of morbidities) are only around 10%[2,3], which makes genetic studies of these traits challenging, as noise tends to obscure effects unless sample sizes are large.

However, with sufficiently large samples, genome-wide association studies (GWAS) of ageing traits have the potential to identify genes and pathways involved in the human ageing process. GWAS have attempted to identify loci and pathways related to healthspan[3,4], (parental) lifespan[5–7] and survival to exceptional old age (often called longevity)[8,9], with some overlap between findings. Multivariate analyses of correlated traits offers the prospect of increased power[10], especially where samples do not overlap, and may be able to identify variants influencing a common underlying ageing process.

Here, we assess the degree of genetic overlap between published GWAS of three different kinds of ageing phenotypes—healthspan, parental lifespan, and longevity (defined as survival to an age above the 90th percentile)—and perform a multivariate meta-analysis to identify genetic variants related to healthy ageing. We subsequently characterise the sex- and age-specific effects of loci which affect all three ageing traits and look up reported associations with age-related phenotypes and diseases. Finally, we link the observed signal in these loci to the expression of specific genes, including some that are currently studied in model organisms, and identify pathways involved in healthy ageing.

## Results

**Genetic correlations between ageing traits**. We explored three public, European-ancestry GWAS of overlapping ageing traits: healthspan ($N = 300,477$ individuals, 28.3% no longer healthy), parental lifespan ($N = 1,012,240$ parents, 60% deceased), and longevity ($N_{cases} = 11,262$; $N_{controls} = 25,483$). The traits show substantial genetic correlations ($P < 5 \times 10^{-8}$) despite differences in age demographics, trait definition, and study design. Parental lifespan correlates strongly with both healthspan ($r_g = 0.70$; SE = 0.04) and longevity ($r_g = 0.81$; SE = 0.08), while healthspan and longevity show a weaker correlation with each other ($r_g = 0.51$; SE = 0.09) (Fig. 1a). We performed an age-stratified GWAS of parental lifespan in UK Biobank to assess whether the genetic correlations between the traits are age-dependent, but our results showed no clear trend in the correlations between healthspan/ longevity and age-stratified parental lifespan bands (Fig. 1b).

We next tested whether differences in ageing trait genetics could be explained by differences in genetic correlations with 27 other traits (Supplementary Table 1). We find all three ageing traits show similar correlations ($P < 0.05/81$; $P_{het} > 0.05$) with coronary artery disease (range healthspan $r_g$ −0.69; SE = 0.07 to parental lifespan $r_g$ −0.49; SE = 0.10), stroke (range parental lifespan $r_g$ −0.56; SE = 0.11 to healthspan $r_g = -0.47$; SE = 0.06), chronic obstructive pulmonary disease (range healthspan $r_g = -0.45$; SE = 0.04 to parental lifespan $r_g = -0.26$; SE = 0.07), and years of schooling (range longevity $r_g$ 0.24; SE = 0.04 to healthspan $r_g$ 0.34; SE = 0.03). However, we also find evidence for differences in correlations across the traits ($P_{het} < 0.05$): healthspan correlated more strongly with metabolic traits (such as

type 2 diabetes) than the other studies, and showed negative genetic correlations with depression and cancers, especially melanoma ($r_g = -0.25$; SE = 0.05), which were not observed in the other datasets. Conversely, parental lifespan correlated uniquely with alcohol intake ($r_g = -0.18$; SE = 0.06) and longevity showed a unique correlation with Alzheimer's disease ($r_g = -0.43$; SE = 0.11). (Fig. 1c; Supplementary Data 1).

**Genome-wide multivariate meta-analysis**. Given the correlations amongst the traits, a combined MANOVA offers the prospect of increased power. We therefore performed a meta-analysis of GWAS of healthspan, parental lifespan, and longevity, which identified 24 loci at genome-wide significance ($P < 5 \times 10^{-8}$) (Fig. 2; Supplementary Data 2; Full summary statistics at https:// doi.org/10.7488/ds/2793). The combined statistics had an LD-score regression intercept of 1.064 (SE 0.009). This suggests limited inflation due to population stratification or relatedness and, in line with some previous studies[11,12], we did not adjust our statistics for this intercept. The APOE locus contained the most significant multivariate SNP ($P < 1 \times 10^{-126}$), associated with an average increase in parental lifespan of 12.7 months per allele (95% CI: 11.4–14.0) and an increased odds ratio of reaching longevity of 1.66 (1.56–1.77). However, noting that <2% of the healthspan study sample experienced Alzheimer's disease, the same allele was associated with an average healthspan increase of only around 50 days (2–98).

Twenty one of the 24 multivariate GWAS loci reaching genome-wide significance had directionally consistent effects in the three studied datasets and 18 were nominally significant ($P < 0.05$) in two or more datasets (Supplementary Fig. 1). A look-up of the lead SNPs and close proxies in the GWAS catalog and PhenoScanner showed that healthspan-specific loci (i.e. $P < 0.05$ only in the healthspan dataset) were mostly associated with skin cancers and metabolic traits, while parental lifespan-specific loci were associated with smoking and risk taking (Supplementary Data 3). Associations with these phenotypes suggests these variants influence (behaviours leading to) environmental exposures and thus likely capture extrinsic ageing processes. As we were primarily interested in genetic variation influencing the intrinsic ageing process, we focused the remainder of this study on genetic variants reaching nominal significance in all three datasets, which are less likely to be associated to study- or population-specific exposures.

Ten loci reached nominal significance ($P < 0.05$) in all three GWAS datasets (Table 1). Five of these are of particular interest as they contain no genome-wide significant SNPs in any individual dataset. The lead multivariate SNP of these loci include rs2643826 (nearest gene *SLC4A7*), rs17499404 (*LINC02513*), rs1159806 (*FOXO3*), rs61905747 (*ZW10*), and rs12830425 (*FGD6*) (Supplementary Figs. 2–6). The lead SNP near *FOXO3* is in moderate linkage disequilibrium (LD) ($r^2 > 0.4$) with rs2802292, a well-known candidate SNP from longevity studies[13]. Given that some of the loci show P values near the genome-wide significance threshold (i.e. *SLC4A7* and *LINC02513*), replication of these loci in large independent cohorts, which were not yet available to us, is warranted.

**Links with sex, age and age-related disease**. We next tested whether loci of interest displayed varying effects on parental lifespan by sex, using sex-specific parental GWAS summary statistics from Timmers et al.[7]. We find evidence of sexual dimorphism for the ApoE ε4 allele ($\beta_{fathers} = 0.08$, $\beta_{mothers} = 0.13$, $P_{diff} = 1.5 \times 10^{-6}$) and evidence of no sexual dimorphism for lead SNPs near *LINC02513*, *SLC4A7*, *LPA*, *TOX3*, and *FOXO3* (<20% difference or $P_{diff} > 0.50$). For the remaining loci, near *CDKN2B-AS1*, *ZW10*, *FGD6* and *LDLR*, effect size point estimates may differ by more

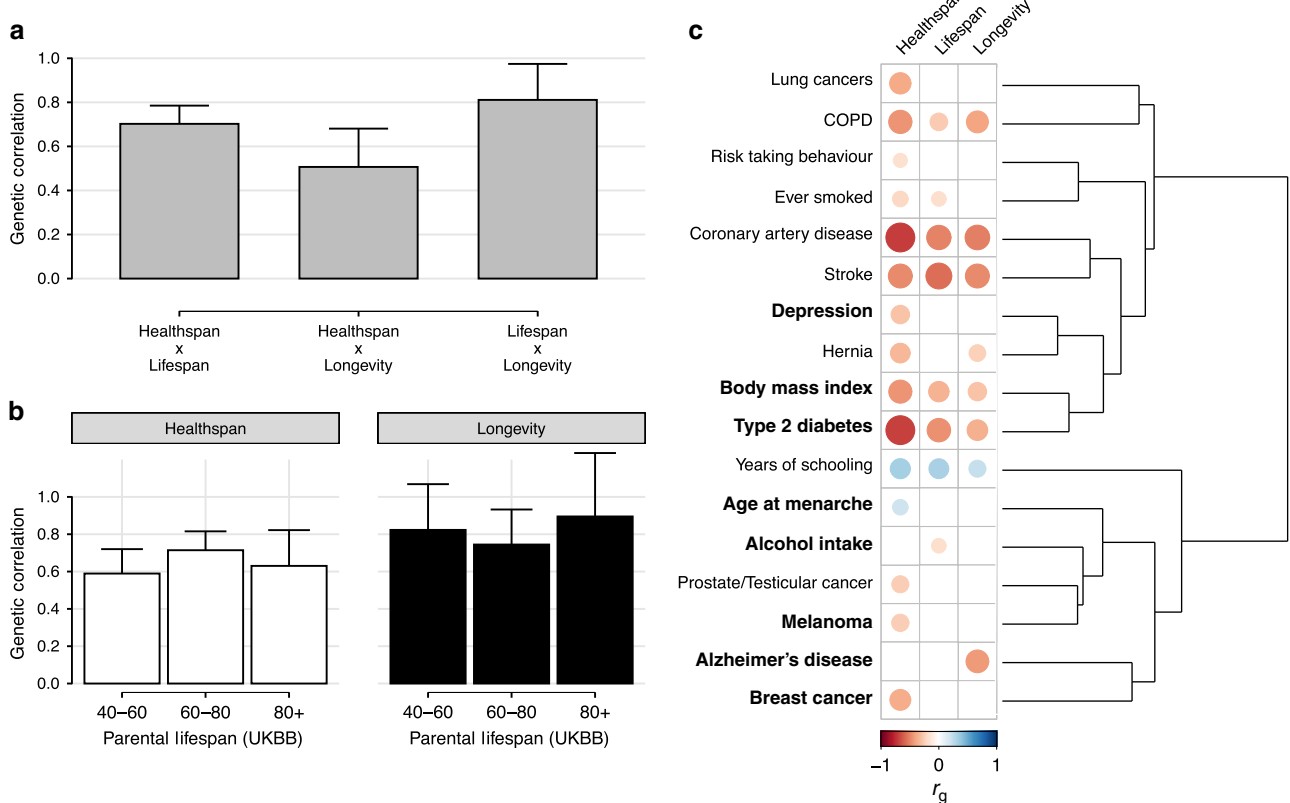

**Fig. 1 Healthspan, parental lifespan, and longevity are highly genetically correlated. a** Pairwise genetic correlation between human ageing studies. **b** Genetic correlations of age-stratified parental lifespan against healthspan and longevity. **c** Genetic correlations ($r_g$) of ageing traits with traits related to development, behaviour, and disease. In bold are traits with heterogeneous correlations ($P_{het} < 0.05$). Displayed here are 17 traits which have at least one significant (FDR < 5%) genetic correlation with healthspan, parental lifespan, or longevity, out of the 27 traits tested. The 17 traits are clustered by Euclidean distance based on their genetic correlation with all tested traits (30 in total). See Supplementary Data 1 for a full list of correlations and Supplementary Table 1 for the number of SNPs used to calculate each pairwise correlation. Blank squares represent correlations which did not pass multiple testing correction. Note that fewer correlations with longevity will pass this threshold due to the smaller sample size of this GWAS. Error bars represent 95% confidence intervals of the correlation estimates. COPD: chronic obstructive pulmonary disease.

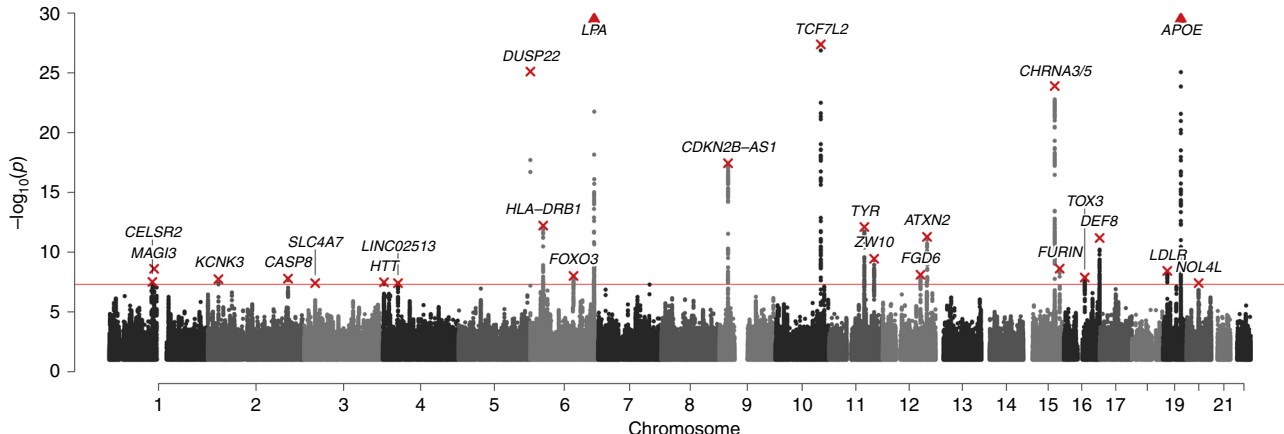

**Fig. 2 Twenty-four multivariate loci identified at genome-wide significance.** Manhattan plot showing the nominal strength of association $-\log_{10}(P$ value) (two-sided) on the *y*-axis against the chromosomal position of SNPs on the *x*-axis, where the null hypothesis is no association with healthspan, parental lifespan, and longevity. The red line represents the genome-wide significance threshold ($5 \times 10^{-8}$). Annotated are the nearest gene(s) to the lead SNP (in red) of each locus. The *y*-axis has been capped at $5 \times 10^{-30}$ to aid legibility; SNPs passing this cap are represented as triangles: *LPA*, $P = 3.8 \times 10^{-30}$, *APOE*, $P = 9.6 \times 10^{-127}$.

**Table 1 Ten loci act across all three ageing traits, reaching nominal significance in each dataset.**

| Nearest Gene | rsID | A1 | Freq1 | $\beta_{healthspan}$ | $P_{healthspan}$ | $\beta_{lifespan}$ | $P_{lifespan}$ | $\beta_{longevity}$ | $P_{longevity}$ | $P_{MANOVA}$ |
|---|---|---|---|---|---|---|---|---|---|---|
| **SLC4A7** | **rs2643826** | **C** | **0.55** | **0.021 (0.005)** | **2E−05** | **0.017 (0.004)** | **2E−05** | **0.045 (0.020)** | **3E−02** | **4E−08** |
| **LINC02513** | **rs17499404** | **A** | **0.54** | **0.017 (0.005)** | **7E−04** | **0.012 (0.004)** | **2E−03** | **0.084 (0.019)** | **1E−05** | **4E−08** |
| **FOXO3** | **rs1159806** | **T** | **0.35** | **0.014 (0.005)** | **5E−03** | **0.015 (0.004)** | **2E−04** | **0.095 (0.020)** | **3E−06** | **1E−08** |
| **ZW10** | **rs61905747** | **A** | **0.82** | **0.029 (0.006)** | **2E−06** | **0.024 (0.005)** | **2E−06** | **0.066 (0.026)** | **1E−02** | **4E−10** |
| **FGD6** | **rs12830425** | **G** | **0.07** | **0.044 (0.009)** | **3E−06** | **0.032 (0.007)** | **2E−05** | **0.077 (0.036)** | **3E−02** | **8E−09** |
| LPA | rs10455872 | A | 0.93 | 0.057 (0.009) | 1E−10 | 0.076 (0.007) | 9E−25 | 0.124 (0.045) | 7E−03 | 4E−30 |
| CDKN2B-AS1 | rs7859727 | C | 0.51 | 0.031 (0.005) | 3E−10 | 0.025 (0.004) | 1E−10 | 0.066 (0.019) | 6E−04 | 4E−18 |
| TOX3 | rs4783780 | A | 0.53 | 0.023 (0.005) | 2E−06 | 0.014 (0.004) | 3E−04 | 0.052 (0.019) | 6E−03 | 1E−08 |
| LDLR | rs6511720 | T | 0.12 | 0.015 (0.007) | 4E−02 | 0.034 (0.006) | 2E−08 | 0.093 (0.030) | 2E−03 | 4E−09 |
| APOE | rs429358 | T | 0.85 | 0.014 (0.007) | 4E−02 | 0.106 (0.005) | 3E−83 | 0.510 (0.032) | 1E−56 | 1E−126 |

$\beta$ effect size of the A1 allele with the standard error in parentheses.
For healthspan and parental lifespan units are the negative log of the hazard ratio, for longevity this is the log odds of reaching an exceptional old age (90th percentile).
In bold are loci which contain SNPs that are not reported at genome-wide significance in any individual dataset (Supplementary Data 4). The remaining loci contain one or more genome-wide significant SNPs within 500 kb of the lead SNP in one of the individual datasets (Supplementary Data 4).
Nearest gene: gene closest to the index SNP, rsID the SNP with the lowest P value in the multivariate analysis, A1 the effect allele, increasing healthspan, parental lifespan, and odds to become long-lived, Freq1 frequency of the A1 allele. P P value of the trait association. For MANOVA, this is the P value against the null hypothesis of association with neither healthspan, parental lifespan, nor longevity.

than 20%, but we would need a larger sample size to be able to detect this difference with confidence (Supplementary Fig. 7).

Looking up the same SNPs in our age-stratified parental lifespan GWAS, we find that all loci, except APOE and SLC4A7, show a downward trend in effect size with parental age. This trend is significant for the APOE locus ($P_{adjusted} = 0.01$), with the effect size of the ε4 allele increasing by 32% (25–39%) for every 10-year increase in parental survival. While we are underpowered to confirm the trends for the remaining loci, we find that, collectively, the average effect of the protective alleles of these nine loci decreases by 24% (13–34%; $P_{adjusted} = 1 \times 10^{-4}$) for every 10-year increase in parental survival (Supplementary Fig. 8).

We also found that the loci of interest had previously been associated at a genome-wide significant level with several age-related diseases and/or phenotypes. The life-extending allele of the majority of the loci is associated with a reduction in cardiovascular disease phenotypes, including SNPs near the ageing loci SLC4A7, FGD6, and LINC02513 discovered in this study. Interestingly, protective variants near FOXO3 are associated with a reduction in metabolic syndrome, but also with a reduction in cognitive ability. Life-extending SNPs near APOE, FOXO3 and FGD6 are all associated with increased measures of macular degeneration (Supplementary Fig. 9; Supplementary Data 3).

**Ageing genes and pathways.** Assessing the loci of interest for colocalisation with gene expression quantitative trait loci (eQTL), we find strong evidence (FDR$_{SMR}$ < 5%; $P_{HEIDI}$ > 1%; see "Methods") of cis-acting eQTL colocalisation for eight out of 10 loci. In total, we highlight 27 unique genes acting across 32 tissues, especially whole blood (12 genes) and the tibial nerve (7 genes) (Supplementary Data 5). In blood, higher expression levels of BCL3 and CKM (near APOE); CTC-510F12.2, ILF3, KANK2 and PDE4A (near LDLR); USP28 and ANKK1 (near ZW10); and CDKN2B are linked to an increase in multivariate ageing traits (i.e. improved survival), while the opposite is true for EXOC3L2 (near APOE), TTC12 (near ZW10), and FOXO3. For the multivariate signal near SLC4A7 we find colocalisation with expression of NEK10 (liver); for the signal near LPA we find colocalisation with expression of SLC22A1/A3 (multiple tissues) and MAP3K4 (pituitary); and for the signal near FGD6 we find colocalisation with expression of FGD6 itself (adipose/arterial). Including trans-acting eQTL from blood, while keeping the same thresholds for colocalisation, we additionally discover higher expression levels of FOXO3B colocalises with the life-extending

signal near FOXO3. When we include genes which could not be tested for heterogeneity ($N_{eQTL} < 3$), we identify one additional cis-acting and 49 additional trans-acting genes (of which 10 colocalise with the signal near LINC02513) (Table 2; Supplementary Data 5).

To determine the age-related expression of the identified cis- and trans-acting genes, we performed a look-up in the dataset of Peters et al.[14]. This large dataset contains the associations of genes with age in whole blood, so we limited ourselves to the cis- and trans-acting genes identified in the whole-blood datasets. We found that FOXO3 expression is increased with age in this dataset, which is in line with the life-extending variant decreasing expression (Supplementary Data 6). Moreover, one cis- (ILF3) and two trans-acting genes (E2F2 and PDZK1IP1) in the LDLR locus show a similar effect (i.e. increased or decreased expression with age combined with the life-extending variant decreasing or increasing expression, respectively). The most interesting, however, seems to be the LINC02513 locus, which showed multiple trans-acting genes to be strongly downregulated with age, while the lead life-extending variant increases expression. LEF1, CCR7, and ABLIM1 even belong to the most significantly affected genes in the whole transcriptomic dataset. This indicates that this long intergenic non-protein coding RNA may serve as a master regulator of age-related transcription in whole blood.

Finally, testing the full list of cis- and trans-acting genes for gene set enrichment in 50 hallmark and 7350 biological process pathways, we find significant enrichment ($P_{adjusted} < 0.05$) in seven hallmark gene sets and 32 biological processes. The hallmark gene sets with the strongest enrichment include haem metabolism, hypoxia, and early oestrogen response (Fig. 3). Enriched biological pathways cluster into categories involving apoptotic signalling, chemical homeostasis, and development of erythrocytes and myeloid cells, among others (Supplementary Fig. 10; Supplementary Data 7).

**Mendelian randomisation of iron traits.** We hypothesised that the effect of haem metabolism and chemical homeostasis on healthspan, parental lifespan, and longevity may be mediated through the bioavailability of iron and investigated this hypothesis using MR of GWAS summary statistics of iron-related traits, i.e., serum iron, log ferritin, and transferrin (percentage saturation and absolute levels), against our GWAS results. In a univariate MR framework, we find evidence of a causal effect for serum iron (FDR < 5%), which appears to be consistent with the MR assumptions and is robust to outliers (Supplementary Fig. 11;

**Table 2 eQTL for 78 genes colocalise with the GWAS signal at 9 out of 10 loci of interest.**

| Locus | Chr | Position | *Cis*-genes | *Trans*-genes |
|---|---|---|---|---|
| SLC4A7 | 3 | 27562988 | NEK10− | |
| LINC02513 | 4 | 38385479 | | EDAR+, MAL+, NOSIP+, CCR7+, ABLIM1+, KRT72+, FHIT+, MMP28+, EPHX2+, LEF1+ |
| FOXO3 | 6 | 109006838 | LINC00222−, FOXO3− | FOXO3B+, MEGF6+, CALCOCO1+, CYBRD1+, IGF1R+, PHF21A+, NDRG1+, KIAA1324−, FCHO2+, CNNM3+ |
| LPA | 6 | 161010118 | SLC22A1+, SLC22A3−, AL591069.1−, MAP3K4− | |
| CDKN2B-AS1 | 9 | 22102165 | CDKN2B+ | |
| ZW10 | 11 | 113639842 | USP28+, ANKK1+, TTC12−, RP11−159N11.4−, ANKK1− | |
| FGD6 | 12 | 95580818 | RP11−256L6.3+, FGD6− | |
| LDLR | 19 | 11202306 | CTC−510F12.2+, KANK2+, SPC24+, SLC44A2+, ILF3+, ILF3−AS1−, DOCK6−, SMARCA4−, PDE4A+ | AHSP−, SELENBP1−, EPB42−, SLC4A1−, HBD−, CA1−, FAM46C−, BLVRB−, TMOD1−, GYPB−, UBE2O−, BPGM−, TRIM58−, SNCA−, IFIT1B−, FECH−, GMPR−, EPB49−, RBM38−, TNS1−, MICAL2−, DCAF12−, RAB3IL1−, PDZK1IP1−, HBM−, BCL2L1−, PLEK2−, E2F2−, TGM2− |
| APOE | 19 | 45411941 | EXOC3L2−, AC006126.4+, CKM+, BCL3+, PVRL2+ | LDLR− |

Genes which showed a significant effect (FDR < 5%) of gene expression on ageing traits are displayed here.
Gene names are annotated with the direction of effect, where + and − indicate whether the life-extending association of the locus is linked with higher or lower gene expression, respectively.
Locus: nearest gene to lead variant in the multivariate analysis, Chr: chromosome, Position: base-pair position of lead variant (GRCh37), *Cis*-genes: genes in physical proximity (<500 kb) to the lead variant of the locus which colocalise with the multivariate signal, *Trans*-genes: genes located more than 500 kb from the lead variant of the locus.

Supplementary Tables 2 and 3). We also find some evidence for an effect of transferrin saturation. However, this association is primarily driven by the well-known hereditary haemochromatosis locus and shows evidence of violating the pleiotropy assumption (i.e. non-zero MR-Egger intercept). We therefore tested all iron traits as exposures simultaneously in a multivariate MR analysis, with our GWAS as the outcome, finding more reliable evidence for causal effects (FDR < 5%; $\beta_{intercept} = 0.0012$; 95% CI −0.0005 to 0.0029) of serum iron, transferrin levels, and transferrin saturation. These effects are not driven by only one of the loci, including the hereditary haemochromatosis locus, as confirmed by leave-one-out analyses (Supplementary Data 8). Although the units of the causal effects are consistent across exposures (and pertinent for *P* values), they are difficult to interpret. We therefore repeated the procedure for the individual component traits: healthspan, parental lifespan, and longevity, recognising the reduction in effective sample size was likely to yield underpowered effect size estimates, although these give a sense of direction and magnitude of the effect in measurable units (Table 3). The multivariate MR effect sizes appear larger than those of the univariate MR, likely due to homeostasis, i.e., variation in one exposure is normally buffered by another. For example, oxidative damage from serum iron may largely be prevented when the metal is bound to transferrin.

## Discussion

Genetic correlations between publicly available healthspan, parental lifespan, and longevity GWAS reveal these traits share 50% or more of their underlying genetics. Performing a multivariate meta-analysis on the GWAS summary statistics, we highlight 24 genomic regions influencing one or more of the traits. Ten regions are of particular interest as they associate with all three ageing traits and are as such likely candidates to capture intrinsic ageing processes, rather than extrinsic sources of ageing. Five of the loci of interest are not associated at a genome-wide significant level in any individual dataset, including the region near FOXO3, which has thus far only been highlighted in candidate gene association studies (reviewed in Sanese et al.[15]) and never at genome-wide significance. The effects of loci of interest on male

and female lifespan are largely the same, although their effect on survival may be slightly stronger in middle age (40–60 years) compared to old age (>80 years). The ApoE ε4 allele is exceptional in this regard, as its effect is stronger in females and increases with age, likely due to its well-known link to Alzheimer's disease[16]. We find these loci of interest colocalise with the expression of 28 *cis*-genes and 50 *trans*-genes, some of which are known to become differentially expressed with increasing age. Lastly, we find these genes are enriched for seven hallmark gene sets (particularly haem metabolism) and 32 largely overlapping biological pathways (including apoptosis and homeostasis), and in line with the highlighted pathways, we find a causal role for iron levels in healthy life in a MR framework.

Interpretation of MR results should be treated with some caution and transparency of the applied method as well as a sensitivity analysis are necessary[17]. In summary, we used an inverse-variance weighted approach in a two-sample MR setting using independent GWAS summary statistics to provide corroboration for the haem metabolism finding (see Methods). The sample overlap between iron-related GWAS and our study was <2.5%. The instrumental variables were independent genome-wide significant SNPs ($P < 5 \times 10^{-8}$), supported by knowledge of biological plausibility as they included several iron transporters and the hereditary haemochromatosis locus, the latter of which had the greatest effect on iron, in line with expectations[18]. The causal effect of iron is further supported by two sensitivity analyses: one showing no evidence of pleiotropy (MR-Egger) and the other showing the observed effect is robust to exclusion of outliers (leave-one-out).

The antagonistic pleiotropy and hyperfunction theories of ageing predict the presence of genetic variants important for growth and development in early life with deleterious effects towards the end of the reproductive window[19,20]. While we are unable to directly capture the genetic effects on individuals before age 40 due to the study design of our datasets, we found that the life-extending variant near FOXO3 is associated with a delay in the age at menarche and a decrease in intracranial volume and cognitive abilities. It thus appears that there are loci exhibiting antagonistic effects, although we are unable to discern whether this is due to true pleiotropy or due to linkage of causal variants within a region

| | Hallmark gene set | N | P | $P_{adj}$ |
|---|---|---|---|---|
| A | Involved in haem metabolism | 19/198 | $2 \times 10^{-25}$ | $1 \times 10^{-24}$ |
| B | Upregulated during hypoxia | 4/193 | $3 \times 10^{-4}$ | $2 \times 10^{-3}$ |
| C | Oestrogen response (early) | 4/196 | $3 \times 10^{-4}$ | $2 \times 10^{-3}$ |
| D | Upregulated by IL2/STAT5 | 4/196 | $3 \times 10^{-4}$ | $2 \times 10^{-3}$ |
| E | Downregulated by KRAS | 3/190 | $3 \times 10^{-3}$ | 0.0180 |
| F | G2/M cell cycle progression | 3/195 | $3 \times 10^{-3}$ | 0.0198 |
| G | Involved in p53 pathway | 3/197 | $3 \times 10^{-3}$ | 0.0205 |

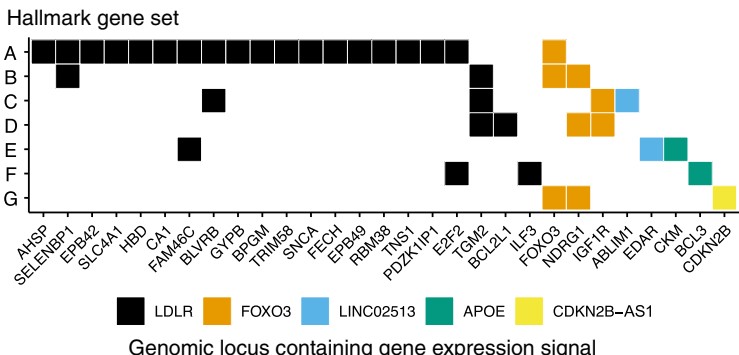

**Fig. 3 Seven hallmark gene pathways are enriched for ageing-related genes.** *N* number of genes of interest vs. total number of genes in the gene set for which eQTL are available. *P* Nominal *P* value of the hypergeometric test for enrichment (against 24,670 background genes). $P_{bonf}$ Bonferroni-corrected *P* value for testing seven hallmark pathways (containing at least three genes). The figure shows individual genes on the *x*-axis and hallmark pathways are listed on the *y*-axis, matching the order of the table. Squares represent the presence of a gene in the gene set.

**Table 3 Multivariate MR of iron-related traits on healthspan, parental lifespan, and longevity shows a protective effect for transferrin and a deleterious effect for serum iron.**

| Exposure | $\beta_{MR}$ | SE | P | $P_{adj}$ | $\beta_{healthspan}$ | $\beta_{lifespan}$ | $\beta_{longevity}$ |
|---|---|---|---|---|---|---|---|
| Serum iron | −0.79 | 0.242 | 1E−03 | 4E−03 | −1.10 (0.58) | −1.17 (0.63) | −5.07 (2.42) |
| Transferrin saturation | 0.80 | 0.252 | 1E−03 | 4E−03 | 1.11 (0.61) | 1.16 (0.66) | 5.15 (2.52) |
| Transferrin | 0.32 | 0.100 | 2E−03 | 4E−03 | 0.48 (0.24) | 0.46 (0.26) | 2.02 (1.00) |
| Ferritin | −0.01 | 0.024 | 0.5380 | 1.0000 | 0.13 (0.06) | −0.02 (0.06) | −0.26 (0.24) |

The effects of 15 SNPs genome-wide significant for one or more iron-related traits were tested against the effects of our GWAS meta-analysis and individual healthspan, parental lifespan, and longevity GWAS in an inverse variance-weighted regression.
Coefficients are derived from a model with a fixed regression intercept, as a sensitivity analysis showed a non-significant intercept centred around zero for all traits ($P_{intercept} \geq 0.76$). Although the causal effect sizes appear large, in practice, homeostatic effects prevent large variation in one of the exposures independent of the others.
$\beta_{MR}$: the causal effect of one standard deviation increase in the exposure on the healthspan/parental lifespan/longevity meta-analysis (in standard deviation units), conditional on the other exposures, *P*: nominal *P* value for the MR effect, $P_{adj}$: multiple testing-corrected *P* value, $\beta_{healthspan}$, $\beta_{lifespan}$, $\beta_{longevity}$: the conditional effect of one standard deviation increase in the exposure on healthspan (in -logHR units), parental lifespan (in -logHR units), or longevity (in logOR units), with the standard error reported in parentheses.

of LD. Should the former be true, this would add to existing evidence for antagonistic pleiotropy in humans, which includes two recent studies that showed antagonistic pleiotropic effects for genes involved in coronary artery disease[21] and ageing[22]. However, almost all loci of interest associate strongly with cardiovascular and blood cell phenotypes, without apparent antagonistic effects, in line with established knowledge that cardiovascular disease is a leading cause of mortality and morbidity worldwide[23].

Several of the genes we identify have previously been shown to influence lifespan in experiments on model organisms. For example, knockouts of the orthologs of *APOE*, *LDLR*, *CDKN2B*, and *RBM38* in mice shortens their lifespan[24–27], while knockout of *IGF1R* has the opposite effect[28]. Similarly, overexpression of the *FOXO3* orthologue in *Drosophila melanogaster*[29] and the *SNCA* orthologue in *Caenorhabditis elegans*[30] have shown to extend their respective lifespans. Many of our genes are also enriched for pathways previously related to ageing in eukaryotic model organisms, including genomic stability, cellular senescence, and nutrient sensing[31]. For example, *FOXO3* and *IGF1R* are well-known players modulating survival in response to dietary restriction[32], but we also highlight genes involved in the response to DNA damage and apoptosis, such as *CDKN2B*, *USP28*, *E2F2*, and *BCL3*. In addition to hallmarks discovered in model organisms, our results suggest that haem metabolism may play a role in human ageing. This pathway includes genes involved in processing haem and differentiation of erythroblasts[33]. Although the enrichment is largely driven by genes linked to the *LDLR* locus, genes linked to other loci

of interest (such as *FOXO3*, *CDKN2B*, *LINC02513*) are involved in similar biological pathways: myeloid differentiation, erythrocyte homeostasis, and chemical homeostasis.

The pathway analysis has potential limitations due to the correlative nature of the genes used to test for enrichment, which can inflate type 1 errors[34]. However, the strong signal for haem metabolism, in combination with the MR results, suggests the evidence for the involvement of this pathway in human ageing is reasonably robust. Haem synthesis declines with age and its deficiency leads to iron accumulation, oxidative stress, and mitochondrial dysfunction[35]. In turn, iron accumulation helps pathogens to sustain an infection[36], which is in line with the known increase in infection susceptibility with age[37]. In the brain, abnormal iron homeostasis is commonly seen in neurodegenerative diseases such as Alzheimer's and Parkinson's disease and multiple sclerosis[38]. Plasma ferritin concentration, a proxy for iron accumulation when unadjusted for plasma iron levels, has been associated with premature mortality in observational studies[39], and has been linked to liver disease, osteoarthritis, and systemic inflammation in MR studies[40,41].

A particular strength of this study is the ability to identify loci shared by multiple traits, without the need for additional sample collection. Comparing the strength of the multivariate association at our 10 loci of interest with the strength of association within each individual GWAS, we estimate the combined statistics are equivalent to a median sample size increase of 127% (95% CI: 52–728%; ~380,876 individuals) for the healthspan study, 76%

(23–146%; ~768,578 parents) for the parental lifespan study, and 415% (59–620%; ~46,726 cases) for the longevity study. This gain in power is particularly important for the latter since the sample size of GWAS for longevity will likely not improve in the near future due to limited availability of data on long-lived people. Having demonstrated the advantages of jointly studying three ageing traits, we encourage future studies to incorporate additional large-scale age-related trait GWAS, such as a recent study on frailty in UK Biobank[42], to further improve power.

It is clear from the association of age-related diseases and the well-known ageing loci APOE and FOXO3 that we are capturing the human ageing process to some extent. However, some judgement is involved in definitions. For one, there are currently no widely accepted standards for measuring healthspan[43]. Zenin et al.[3] define healthspan based on the incidence of the eight most common diseases increasing exponentially in incidence with age in their sample. As such, their trait is highly dependent on the characteristics of the UK Biobank cohort, who were aged 40–69 years when they were recruited in 2006–2010 and of which two-thirds have yet to experience an age-related disease. Therefore, loci with effects on diseases of middle age (cancer and heart disease) are likely overrepresented in our analysis. The rarity of Alzheimer's disease in the UK Biobank sample also explains the limited association of APOE in the healthspan GWAS, compared to the other ageing traits. Similarly, the parental lifespan GWAS is dependent on the most common causes of death in the parental generation. As such, the observed cardiometabolic associations may, to some extent, reflect the large effect of these diseases on death in Europe a few decades ago. Future studies on additional cohorts with wider age ranges, disease frequencies, and causes of death (including individuals of non-European ancestries) would be able to show if loci shared between ageing traits are indeed independent from cohort-specific characteristics and reflect common biological ageing mechanisms.

Multivariate analysis of traits does not provide a natural combined effect size or direction of effect. Colocalisation of eQTL with loci of interest requires effect directions to test for heterogeneity of instruments. As such, we used the direction of the sum of the $Z$ scores of the underlying traits to assign a direction to $Z$ scores derived from MANOVA $P$ values. This works well for SNPs with concordant effects on ageing traits but is less accurate when SNPs have heterogeneous or antagonistic effects. For example, a SNP associated with an increase in healthspan and an equal decrease in parental lifespan—while likely rare—will have a large $Z$ score in the MANOVA, but no clear direction of effect. This limitation will introduce some heterogeneity in the colocalisation analysis, and as a result inflate the HEIDI statistic. Furthermore, gene expression colocalisation is limited by the number of tissue eQTL (with some tissues being underpowered) and does not capture the effect of coding variation. There may be additional genes with highly tissue-specific effects or effects dependent on structure or splicing isoforms, which we are unable to detect.

The pathways we have highlighted mostly relate to biological processes for chemical and cellular homeostasis and are therefore likely to be generalisable across populations. However, it is important to note that all GWAS summary statistics used in our study were derived from individuals of European ancestry and more follow-up work is necessary to validate our findings in individuals from other ethnic backgrounds. For example, certain population characteristics, such as levels of obesity and meat intake, can affect the bioavailability of iron[44] and thus the relative importance of haem metabolism in ageing.

Another limitation is that our meta-analysis, like many others, is focused on the identification of additive genetic variants. The evolutionary theory of ageing predicts that recessive variants may have larger effects on fitness[45], and this prediction is supported by

a recent study on the relation between recessive mutations involved in haemochromatosis and morbidity[41]. Heritability studies of lifespan show that a small but significant amount of phenotypic variation may be explained by dominance effects[46], and, as such, future studies should also try to study the effect of recessive variants on ageing traits.

Importantly, the genes we have highlighted show natural variation in the human population and some of them show altered levels of expression with increasing age, which makes them good candidates for therapeutic intervention. However, colocalisation of gene expression could be due to pleiotropy rather than causality, and there is a need to validate the effects of genetic variants in experimental models to confirm their role in disease aetiology. For example, we have found life-extending variants colocalise with decreased expression of FOXO3 in blood, which itself becomes increasingly expressed with increasing age, but experiments suggest the gene has many protective functions including detoxification of reactive oxygen species and DNA damage repair[15]. The observed inverse relationship between healthy life and FOXO3 expression may reflect healthy individuals have less oxidative damage and require less FOXO3 to mitigate this damage.

In conclusion, the challenge of studying ageing genetics in humans—low heritability and limited samples—can be overcome to some extent by combining large studies of closely related phenotypes that capture elements of the ageing process. Focusing on the overlap between different populations and age-related traits has revealed that several ageing pathways discovered in model organisms also apply to humans, and has highlighted genes and pathways in humans which can now be further tested in model organisms. This study, and follow-up work on the genes we have highlighted, will eventually lead to therapeutic targets that can reduce the burden of age-related diseases, extend the healthy years of life, and increase the chances of becoming long-lived without long periods of morbidity.

## Methods

**Data sources**. We downloaded three publicly available sets of summary statistics on healthspan[47] (https://doi.org/10.5281/zenodo.1302861), parental lifespan[48] (https://doi.org/10.7488/ds/2463), and longevity[9] (https://www.longevitygenomics.org/downloads), whose derivation is briefly described here.

*Healthspan*. The Healthspan GWAS consists of 300,477 unrelated, British-ancestry individuals from UK Biobank. The statistics were calculated by fitting Cox-Gompertz survival models with events defined as the first incidence of one of seven diseases (any cancer, diabetes, myocardial infarction, stroke, chronic obstructive pulmonary disease, dementia, and congestive heart failure) or death itself. Martingale residuals from this model were then regressed against HRC-imputed dosages. Of the 84,949 individuals who had experienced an event (and thus had complete healthspans), 51.3% experienced a cancer event, 18.0% a diagnosis of diabetes and 17.1% a myocardial event. Less than 5% of the individuals experienced their first event due to one of the remaining diseases. See Zenin et al.[3] for details. After removing single nucleotide polymorphisms (SNPs) with duplicate rsIDs ($N = 19,386$) summary statistics were available for 5,429,268 common (MAF ≥ 0.05) and 5,860,562 rare (MAF < 0.05) SNPs.

*Parental lifespan*. The Parental Lifespan GWAS consists of unrelated, European-ancestry individuals reporting a total of 512,047 mother and 500,193 father lifespans, of which 60% were complete. The statistics for each participating cohort were calculated by fitting Cox survival models to father and mother survival separately, adjusted for subject sex, at least 10 principal components, and study-specific covariates such as genotyping batch and array. Martingale residuals of the survival models were regressed against subject dosages (HRC-imputed). Father and mother results were combined into two separate ways: father and mother residuals from UK Biobank were combined before regression, while father and mother summary statistics from other cohorts were meta-analysed, adjusting for the phenotypic correlation between parents. See Timmers et al.[7] for details. Summary statistics were available for 5,526,246 common (MAF ≥ 0.05) and 3,559,402 rare (MAF < 0.05) SNPs.

*Longevity*. The Longevity GWAS consist of unrelated, European-ancestry individuals who lived to an age above the 90th survival percentile ($N_{cases} = 11,262$) or whose age at the last follow-up visit (or age at death) was at or before the 60th

percentile age ($N_{controls}$ = 25,483). The statistics for each of the participating cohorts were calculated using logistic regression and 1000 G Phase 1 version 3-imputed dosages, adjusted for clinical site, known family relationships, and/or the first four principal components (if applicable) and subsequently combined using a fixed-effect meta-analysis. See Deelen et al.[9] for details. After removing SNPs with duplicate IDs ($N$ = 17,152), summary statistics were available for 6,657,238 common (MAF ≥ 0.05) and 2,181,962 rare (MAF < 0.05) SNPs.

**Age-stratified survival analysis.** We carried out a series of additional, age-stratified GWAS using a sample of 325,614 unrelated, British-ancestry individuals from UK Biobank (as determined by genomic PCA and 3rd degree kinship or closer)[49], in order to calculate age band-specific effects of SNPs on parental lifespan. These individuals answered questions regarding their family history via touchscreen questionnaire, including their adoption status and parental age or age at death if deceased. Quality control was performed in R version 3.6.0 as in Timmers et al.[7], starting with 409,692 British-ancestry individuals and excluding subjects who were adopted, had two parents who died before age 40, or who did not provide information on parental age ($N$ = 12,406; 3.0%). In addition, we excluded individuals who had withdrawn their consent to participate as of 16 October 2018 and all but one of each related set of individuals ($N$ = 71,672; 17.5%). Related individuals were excluded as mixed modelling is not well understood in the context of the kin-cohort method[7]. The remaining 325,614 individuals reported 312,088 and 322,672 father and mother lifespans, respectively, of which 67.7% were complete. Parent lifespans were then split into three age bands, 40–60, 60–80, and 80–120, excluding parents who died before the start of the age band and treating any parent who survived at least until the end of the age band as alive (i.e. right-censored). Sample descriptives of each age band are detailed in Supplementary Table 4. Using the R package survival, Cox proportional hazard models were fitted separately to each father and mother age band—six combinations in total—adjusted for subject sex, genotyping batch and array, and the first 40 genetic principal components.

$$h(x) = h_0(x)e^{\beta_1 \mathbf{X}_1 + \beta_2 \mathbf{X}_2 + ... + \beta_n \mathbf{X}_n}, \qquad (1)$$

where $h(x)$ is the hazard of the parent at age $x$, $h_0(x)$ the baseline hazard, and $\beta_1$, $\beta_2, ..., \beta_n$ the effect sizes (natural log of the hazard ratio) associated with the covariates $\mathbf{X}_1, \mathbf{X}_2, ..., \mathbf{X}_n$. Martingale residuals of these models were taken[50], divided by the proportion dead to scale effects to hazard ratios and doubled to account for parental genotype imputation[5], and then regressed against subject allelic dosage in an additive model using RegScan[51]. Individual parental lifespan statistics were combined using inverse-variance meta-analysis, inflating standard errors by $\sqrt{1 + r_p}$ to take into account the correlation between the parental phenotypes ($r_p$).

**Genetic correlation analysis.** LD-score regression[52] was used to calculate genetic correlations between ageing trait GWAS, age-stratified parental lifespan (described above) and 27 European-ancestry GWAS of developmental, behavioural, and disease traits (Supplementary Table 1). In line with recommendations[53], imperfectly imputed (INFO < 0.9) and low frequency (MAF < 0.05) SNPs, as well as those located in the Major Histocompatibility Complex, were discarded before merging the summary statistics with a HapMap3 reference panel to reduce statistical noise. An average of 1,086,952 SNPs (range 866,405–1,181,238) were used to calculate genetic correlations per set of summary statistics, based on LD-score regression weights derived from European individuals.

**Multivariate genomic scan.** Healthspan, parental lifespan, and longevity summary statistics were meta-analysed using MANOVA, while accounting for correlations between studies due to (limited) sample overlap and correlation amongst the traits, as implemented in MultiABEL v1.1-6[10]. Correlations were calculated from summary statistics by taking the correlation in effect estimates from independent SNPs between studies (60,338 default SNPs provided by MultiABEL and shared between studies). These correlation estimates ranged from 0.013 between healthspan and longevity to 0.094 between healthspan and parental lifespan, reflecting a small degree of sample overlap and/or phenotypic correlation. Summary association statistics were calculated for the 7,320,282 SNPs shared between studies, of which 5,278,109 were common (MAF ≥ 0.05) and 2,042,173 were rare (MAF < 0.05). These statistics represent the significance of each SNP affecting one or more of the traits, giving a $P$ value against the null hypothesis that effect sizes are zero in all studies. The method does not provide a combined effect size.

Loci were defined as 500 kb regions flanking the lead genome-wide significant SNP in linkage equilibrium ($r^2$ < 0.1) with other lead SNPs. LD-score regression was used to assess inflation of the GWAS statistics, using 1,138,687 SNPs from the MANOVA and LD weights from European samples from the 1000 Genomes project. Loci with lead SNPs showing a nominally significant effect ($P$ < 0.05) in all three datasets were considered more likely to capture intrinsic ageing pathways. We refer to them as loci of interest throughout this study.

**Sex- and age-stratified analyses.** Lead SNPs of loci of interest were looked up in individual father and mother survival statistics from Timmers et al.[7]. Differences in the parental effect sizes were tested using $(\beta_{fathers} - \beta_{mothers})/\sqrt{(SE_{fathers}^2 + }$

$SE_{mothers}^2)$, where SE is the standard error of the effect estimate. This statistic follows a $Z$ distribution, assuming errors in measured effects are independent.

Age-specific survival statistics were retrieved for the same loci from our age-stratified parental lifespan GWAS in UK Biobank. In order to standardise effects for each locus, we expressed the age-specific effect as a fold change from the unstratified effect in UK Biobank, inflating standard errors using the Taylor series expansion to account for the uncertainty in the denominator:

$$\alpha = \frac{\beta_{band}}{\beta_{all}} - 1 \qquad (2)$$

$$SE_\alpha = \sqrt{\frac{SE_{band}^2}{\beta_{all}^2} + \frac{\beta_{band}^2 SE_{all}^2}{(\beta_{all}^2)^2}}, \qquad (3)$$

where $\alpha$ is the fold change in effect, $\beta_{band}$ is the effect estimate of the age-specific band, $\beta_{all}$ is the unstratified effect estimate, and SE is the standard error of the respective effects.

This provided a relative change in effect size per parental age band. We then calculated the median survival from Kaplan–Meier survival curves of each age band, allowing us to place the effects on a years-of-life scale. For each locus individually, effect sizes of age bands were regressed against median survival of the age band, inversely weighted by the variance of the effect estimate (constituting 10 statistical tests). Coefficients of the loci underpowered to detect a trend individually ($P$ > 0.05/10) were meta-analysed, again weighted by the inverse of their variance, to provide a collective estimate. A sensitivity analysis examining the collective trend estimate using all loci of interest (instead of only underpowered loci) was performed using the meta R package and found substantial heterogeneity ($I^2$ > 89%) driven by APOE, which represented almost 70% of the regression weights. As such, the $P$ values for age-specific effects reported in the main text were Bonferroni-adjusted for a total of 12 statistical tests.

**Associations with diseases and traits.** Lead SNPs from the multivariate GWAS and close proxies ($r^2_{EUR}$ > 0.6) were looked up in the GWAS catalog[54] and PhenoScanner[55] (accessed 14 October 2019). All genome-wide associations were included except triallelic SNPs, associations without effect sizes, and associations with healthspan, parental lifespan, longevity, or medications. Similar traits were then grouped together using approximate string matching—verified manually—keeping only the strongest association and the shortest trait name. For example, Body mass index, Body mass index in smokers, and Body mass index in females greater than 50 years of age were grouped and renamed to Body mass index. Associations were then categorised into seven disease phenotypes based on keywords and manual curation: Cardiovascular, metabolic, neuropsychiatric, immune-related, smoking-related, cancer, and age-related. Cardiovascular phenotypes included lipid levels, vascular traits, and diseases concerning the heart; Metabolic phenotypes included body (fat) mass and glycaemic traits; Neuropsychiatric phenotypes included neurodegenerative diseases and disorders of brain signalling such as restless leg syndrome and epilepsy; Immune-related phenotypes included measures of immune cells, and inflammatory and autoimmune disorders; Smoking-related phenotypes included smoking and lung function-related traits; Cancer included all neoplasms and carcinomas; Age-related phenotypes included traits typically associated with advancing age, such as age at menopause, male pattern baldness, age-related macular degeneration, hearing loss, and frailty. See Supplementary Data 9 for a list of all phenotypes within each category.

**Gene expression colocalisation analysis.** For each locus of interest, gene expression was tested for colocalisation with SNP effects within 500 kb of the lead SNP using SMR-HEIDI[56,57]. The gene expression studies included Westra (cis-eQTL), CAGE (cis-eQTL), Vosa (cis- and trans-eQTL), and GTEx v7[58-61], the latter with eQTL $P$ < 10^{-5} only. Estimates of SNP effects are needed for SMR but are not directly provided by the multivariate analysis. Instead, we derived Z scores from multivariate $P$ values and signed these based on the sign of the sum of underlying healthspan, parental lifespan, and longevity Z scores. The HEIDI statistic is dependent on the heterogeneity between effect estimates. We therefore recalculated standard errors and effect sizes based on allele frequency and sample size, using formula 6 from Zhu et al.[56]. For sample size, we used the sum of individual studies' effective samples ($N$ = 709,709) and performed a sensitivity test using the sum of all samples (regardless of their contribution to study power; $N$ = 1,349,432). Differences in $P_{HEIDI}$ between analyses were <0.0006, i.e., had no practical effect on results. A Benjamini–Hochberg multiple testing correction was applied separately to each eQTL dataset to account for the number of genes tested. Determining an optimal threshold for heterogeneity pruning is less straightforward: Wu et al.[57] consider 5% to be too conservative, especially when using summary-level data and SNPs with different sample sizes, and set a 1% threshold to correct for three colocalisation tests. We apply the same threshold, which may still be conservative in our study as we test many (albeit partially overlapping) tissues and we expect additional heterogeneity due to inferred Z scores (see Discussion).

**Gene set enrichment analysis.** Genes colocalising with loci of interest in cis or trans at FDR < 5% were tested for enrichment in 50 GO hallmark and 7350

biological process gene sets from the Molecular Signatures Database[33], using a procedure analogous to Gene2Func in FUMA[62]. First, we translated all unique gene symbols from the eQTL datasets to Entrez IDs ($N = 24,670$), and subsetted hallmark and GO biological process gene sets to only include genes for which eQTL were available. We then used a hypergeometric test to assess whether our genes were overrepresented in each pathway compared to all genes with eQTL. A minimum of three genes had to be present in a gene set for it to be tested for enrichment. Seven hallmark gene sets and 383 biological process gene sets met this requirement. Bonferroni correction was applied to account for multiple testing, separately for hallmark and biological process sets. Gene sets with $P_{bonferroni} < 5\%$ are reported.

**Mendelian randomisation analysis**. Summary statistics for serum iron, ferritin, transferrin, and transferrin saturation were obtained from Benyamin et al.[18] to be used as exposures in a Mendelian randomisation (MR) analysis. Univariate MR was performed using the R package TwoSampleMR[63], with instrumental variables defined as the lead genome-wide significant SNPs ($P < 5 \times 10^{-8}$; at least 1 Mb apart) shared between each iron-related trait and our meta-analysis ($N_{iron} = 6$; $N_{ferritin} = 5$; $N_{transferrin} = 11$; $N_{saturation} = 9$). We calculated the effect estimates of our multivariate meta-analysis using the same method as described for SMR-HEIDI. For each iron-related trait, two inverse variance-weighted regressions were run with the iron-related trait as exposure and the inferred multivariate effect sizes as outcome: one without a fixed intercept to test for pleiotropy violation (MR Egger) and one with the intercept set to zero. We adjusted $P$ values for multiple testing using the Benjamini–Hochberg method with an FDR threshold of 5% for significance. We also performed a leave-one-out analysis to assess whether the observed effects were robust to outliers.

Multivariate MR was performed using the R package MendelianRandomization[64], which could fit a random-effects model and provides an estimate of the multivariate regression intercept but is otherwise identical to TwoSampleMR. As instrumental variables, we used the same iron trait-related lead SNPs as before, keeping only the SNP with the strongest association if multiple lead SNPs were located in the same 1 Mb locus ($N = 15$). We fitted a random-effects, inverse variance-weighted model, with and without fixed intercept, with the multivariate ageing trait as outcome, and as before, performed a sensitivity analysis using the leave-one-out method. The main analysis was then repeated using the same SNPs and effects derived from the original healthspan, parental lifespan, and longevity GWAS as outcomes. $P$ values were adjusted for eight tests (four traits, with and without intercept) using the Benjamini–Hochberg method.

**Reporting summary**. Further information on research design is available in the Nature Research Reporting Summary linked to this article.

## Data availability
The healthspan, parental lifespan, and longevity GWAS summary statistics are available from OpenAIRE (https://doi.org/10.5281/zenodo.1302861), Edinburgh DataShare (https://doi.org/10.7488/ds/2463), and the longevity genomics website (https://www.longevitygenomics.org/downloads), respectively. The multivariate GWAS summary statistics generated in this study are available from Edinburgh DataShare with the identifier https://doi.org/10.7488/ds/2793. The various summary statistics used to calculate genetic correlations are available from GeneAtlas (http://geneatlas.roslin.ed.ac.uk/), NealeLab (http://www.nealelab.is/uk-biobank), or their respective publications. The lists of SNP-trait associations are available from the GWAS catalog (https://www.ebi.ac.uk/gwas/) and PhenoScanner (http://www.phenoscanner.medschl.cam.ac.uk/). The hallmark and biological process gene sets are available from the Molecular Signatures Database (https://www.gsea-msigdb.org/). Source data for figures in this study are available in the supplementary documents and upon request from the corresponding authors.

## Code availability
Statistical code is available at https://github.com/PaulTimmers/NCOMMS-20-00614.

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

## Acknowledgements

We would like to thank the authors of the many GWAS used in this work for making their summary statistics publicly available. We would also like to acknowledge funding from the Medical Research Council (P.R.H.J.T.: MR/N013166/1, J.F.W.: MC_UU_00007/10); the University of Edinburgh (P.R.H.J.T. and P.K.J.); and the Alexander von Humboldt Foundation (J.D.).

## Author contributions

P.R.H.J.T.: Conceptualization, Methodology, Software, Validation, Formal Analysis, Investigation, Writing—Original draft preparation, Writing—Review and editing, Visualization. J.F.W: Supervision, Writing—Review and editing. P.K.J.: Conceptualization, Supervision, Project administration, Validation, Writing—Review and editing. J.D.: Conceptualization, Investigation, Writing—Original draft preparation, Writing—Review and editing.

## Competing interests

The authors declare no competing interests.
