## [Peer Review File · Nature Communications]

Reviewers' comments:

Reviewer #1 (Remarks to the Author):

The relationship between healthspan and lifespan is an interesting and important area of study: why some people live a long time in good health where others don't. Understanding the biological mechanisms at play here will inform future research in prediction and treatment. Timmers et al. have performed an analysis of genetic variants associated with healthspan and lifespan (and longevity, the extreme lifespan phenotype) in multivariate analysis to identify the variants (and genes and pathways) associated with healthy ageing. They identify 10 genetic loci reaching nominal significance in all three phenotypes: some well-known (APOE, 9p21, LDLR, and LPA) in cardiovascular disease, among other phenotypes, and others less-so. This is the first time, to my knowledge, the FOXO3 has convincingly appeared at genome-wide significance, despite the pathway being prominent in model organism work, and that the haem pathway has been implicated in healthy ageing. Overall this is an interesting and timely study, combining three analyses of related but different phenotypes to identify shared causes that should be of general interest. I have no major concerns with the approach or conclusions drawn, but have some minor comments:

- I have always had some concerns about the healthspan GWAS because although it includes a number of disease outcomes the results are dominated by the SNPs affecting the most common diseases, as is clearly demonstrated in Figure 1C of this paper (Healthspan GWAS correlates very strongly with cardiometabolic disease). To some degree this is also true of lifespan/longevity: the results are dominated by cardiometabolic SNPs because it is these diseases that kill most people. I realise the authors somewhat discuss this in a sentence in the discussion (end of page 11, lines 395-398) but I think this warrants much more emphasis/recognition: that GWAS in a population with different rates of disease/causes of death would likely find different loci. It is therefore not surprising that cardiovascular loci dominate the results, and confounding these with mechanisms of ageing (not that the authors said this explicitly) is potentially a mistake.
- Cancer is likely less prominent only because each cancer is actually quite distinct, genetically. The hope that loci affecting "biological age" or similar would emerge has (so far) not really come to pass, perhaps with the exception of APOE and 9p21 (CDKN2B-AS1). The evidence for haem synthesis as an ageing pathway is compelling and an interesting novel aspect to this paper, although it is intriguing that the only GWAS locus linked to this pathway is LDLR, which clearly has a CVD-link to healthy ageing. Have the authors also considered the link between iron and infection susceptibility? Ferritin is a store for iron and is raised during infection as iron availability is a key rate-limiting step for some bacteria (i.e. if none is available, proliferation not possible). Infection susceptibility increases with age and this may be part of the story (i.e. decreased haem synthesis > increased iron availability > infection).
- This GWAS, like so many, is limited by the additive nature of the analyses. For example on page 8 when describing the APOE locus "an average increase in lifespan of 12.7 months per allele." Yet surely the a priori for this locus is a non-additive effect, given the known recessive Alzheimer's effect. I wonder what else is missed by this omission. My comment here is not suggesting the authors repeat the 3 published GWAS looking for non-additive effect (although, that would be great to see) but to acknowledge this in the paper and especially conclusions when discussing the low heritability of healthspan/lifespan (much is probably missed by just focussing on additive, mostly common, effects).

Reviewer #2 (Remarks to the Author):

Timmers et al present an investigation of loci affecting ageing in humans. By comparing the results

of multiple association studies into ageing, they determine that three common measures of healthy ageing (healthspan, lifespan, and longevity) share substantial genetics, and are linked to multiple other plausible phenotypes. Given the shared genetics, Timmers et al combine these ageing studies in a meta-analysis to identify multiple loci linked to a healthy / long-lived phenotype, some of which they report for the first time at genome-wide significance. Timmers et al conclude with an investigation into the possible biology underlying these loci, based on eQTL and gene overrepresentation analysis. This paper is an incremental advance on previous work in the field by the authors and others, but nonetheless its increased power over past studies and novel findings of genome-wide significance make it a valuable contribution.

This paper makes heavy use of statistical genetics methodology. This is an area in which I am not an expert, so I have focused my review on the statistical and biological reasoning of the paper. My conclusion is that this is a solid body of work which is publishable if one major concern is addressed.

Major concern

The final section of the paper aims to assign biological mechanism to the loci identified in the earlier sections. The authors do this by first performing an eQTL colocalisation analysis vs their identified loci, and then running a pathway enrichment scan for the eQTL gene hits. This is a valid hypothesis-generating approach, however I do have a significant concern about the exact procedure used for the pathway enrichment analysis: as it is used currently, I believe that there is a high likelihood of false positive results from this procedure. In a similar vein, due to the solely correlative nature of gene enrichment analysis, the discussion of this result would also benefit from a little moderation.

The hypergeometric test employed by the authors relies on two assumptions which I strongly suspect are violated by these data: equal opportunity for eQTL colocalisation tagging across genes, and independence of gene selection. The authors acknowledge the former (lines 452-455) but do not seem to account for it in the test; for more details of the latter see eg <https://doi.org/10.1093/bioinformatics/btm051>. The overall effect of these assumptions being violated is a high likelihood of a false positive result. Given that the results of this analysis lead to a central conclusion of this paper (the association of haem metabolism with ageing), a more careful analysis here is needed, particularly as the haem metabolism finding seems driven almost exclusively by the eQTLs colocalising with a single age-associated locus. The authors should either undertake an analysis that is more robust to these two issues (perhaps employing permutation?), apply an orthogonal approach to confirm the findings of the hypergeometric test, or de-emphasise the pathway association and haem findings, which I see as highly tentative without additional evidence.

Lines 404-410: This wording is too strong given the evidence presented. Even if my technical concern about the hypergeometric test is addressed, gene enrichment analysis provides at most suggestive evidence of a biological link. A reader might take from this section and the title that haem metabolism is important in human ageing, which is a conclusion that in my opinion is not completely supported by the evidence presented. This section should be moderated to make it clear that although the gene enrichment analysis is suitable as a hypothesis generating tool, it is premature to link haem metabolism with ageing on the basis of these results alone.

Minor concerns

Line 115: The citation for Therneau, Grambsch, & Fleming (1990) has suffered a typo.

Line 150: The authors tested gender-specific survival association for lead SNPs by assuming that

paternal and maternal effects are independent. This is a surprising assumption to me, and indeed my reading of Timmers 2019 suggests that many loci have sex-concordant effects. It would be helpful if the authors could explain in a little more detail why the assumption of independence was made.

Lines 159-168: It seems a lot of tests were done here -- at least one per locus, plus the combined test of the loci without a significant trend. To aid interpretation of these results the authors should include details of what multiple testing control was applied.

Lines 304-321: Unfortunately confidence interval lines for Figures S7 and S8 were not visible to me, as such I could not fully review this section.

Lines 386-398: The argument about antagonistic pleiotropy is interesting, but I think that the results of the study are suggestive in this regard, not conclusive. Thus the statement that "it is clear that loci exhibiting antagonistic pleiotropy exist in humans" is a little strong if based on this study alone, and I suggest a moderation of the claims here. The authors could perhaps address this by discussing with more reference to the recent literature, eg [dx.doi.org/10.1038/s41559-016-0055](https://doi.org/10.1038/s41559-016-0055) and [dx.doi.org/10.1371/journal.pgen.1006328](https://doi.org/10.1371/journal.pgen.1006328).

Lines 411-419: In my opinion this paragraph adds little, and strays rather far from the questions posed in the Introduction. Unless a deeper investigation of haem metabolism and ageing is added to the results, I think the paper would be clearer without this paragraph.

Other comments and suggestions

It would be interesting if the authors could link their eQTL findings to the existing body of literature on associations between gene expression and ageing / longevity in humans and model organisms.

Mark Pinese

Reviewer #3 (Remarks to the Author):

This manuscript performed multiple traits analysis of healthspan, lifespan and longevity and identified 10 loci that influencing all three traits by using summary statistics of three large cohorts: the HealthSpan, the Parental Lifespan and the Longevity. Among them, five are novel. There is suggestion of antagonistic pleiotropy. The manuscript was well written. I have some comments.

Major

- 1) The multivariate analysis of the three GWAS summary statistics resulted an inflation parameter 1.064 from LD regression. After adjusting for this inflation parameter, it is apparently that the two novel loci (SLC4A7 and LINCO2513) are no longer genome wide significant. In addition, the P values of the novel loci are all close to $5E-8$. Additional replication analysis is necessary to verify these novel loci.
- 2) The manuscript concluded that there is antagonistic pleiotropy among the loci identified. However, this is just a speculation. There is no clear evidence from the current analysis. Although combining multiple traits will improve statistical power for detecting associated loci, the MANOVA analysis cannot differentiate pleiotropy or colocalization. The evidence is quite weak to claim that a genetic variants are associated with all three traits by restricting the variants with a P value < 0.05

for all three traits.

Minor:

1) In figure 1, how the tree in c) was obtained? The sample size of the Longevity GWAS was the smallest. This can affect the estimation of genetic correlations.

2) Page 4, line 113 and 114, $\beta_{1,2,\dots,n}$ should be $\beta_1, \beta_2, \dots, \beta_n$ and $X_{1,2,\dots,n}$ should be X_1, X_2, \dots, X_n .

We thank the Reviewers for their helpful and insightful suggestions, which we believe have improved the strength and depth of our manuscript, and we hope we have properly addressed them.

Reviewer #1

I have always had some concerns about the healthspan GWAS because although it includes a number of disease outcomes the results are dominated by the SNPs affecting the most common diseases, as is clearly demonstrated in Figure 1C of this paper (Healthspan GWAS correlates very strongly with cardiometabolic disease). To some degree this is also true of lifespan/longevity: the results are dominated by cardiometabolic SNPs because it is these diseases that kill most people. I realise the authors somewhat discuss this in a sentence in the discussion (end of page 11, lines 395-398) but I think this warrants much more emphasis/recognition: that GWAS in a population with different rates of disease/causes of death would likely find different loci. It is therefore not surprising that cardiovascular loci dominate the results, and confounding these with mechanisms of ageing (not that the authors said this explicitly) is potentially a mistake.

It is indeed true that the most significant SNPs in our meta-analysis do influence healthy ageing through cardiovascular-related traits. At the same time, however, our meta-analysis revealed some loci influencing healthy ageing via other routes, showing that we have the power to detect these. We have expanded the Discussion section (lines 527-533) to further emphasise the limitation of population-specific diseases and causes of death as mentioned by the Reviewer.

*Cancer is likely less prominent only because each cancer is actually quite distinct, genetically. The hope that loci affecting “biological age” or similar would emerge has (so far) not really come to pass, perhaps with the exception of APOE and 9p21 (CDKN2B-AS1). The evidence for haem synthesis as an ageing pathway is compelling and an interesting novel aspect to this paper, although it is intriguing that the only GWAS locus linked to this pathway is LDLR, which clearly has a CVD-link to healthy ageing. **Have the authors also considered the link between iron and infection susceptibility?** Ferritin is a store for iron and is raised during infection as iron availability is a key rate-limiting step for some bacteria (i.e. if none is available, proliferation not possible). Infection susceptibility increases with age and this may be part of the story (i.e. decreased haem synthesis > increased iron availability > infection).*

This is an interesting point. Based on the suggestion of the Reviewer, we have amended our Discussion section (lines 495-497) to discuss the link between iron and susceptibility to infection. Moreover, based on the comments of Reviewer #2, we have performed a Mendelian randomisation analysis, which provides additional evidence that iron-related traits play a role in healthy ageing.

This GWAS, like so many, is limited by the additive nature of the analyses. For example on page 8 when describing the APOE locus “an average increase in lifespan of 12.7 months per allele.” Yet surely the a priori for this locus is a non-additive effect, given the known recessive Alzheimer’s effect. I wonder what else is missed by this omission. My comment here

is not suggesting the authors repeat the 3 published GWAS looking for non-additive effect (although, that would be great to see) but to acknowledge this in the paper and especially conclusions when discussing the low heritability of healthspan/lifespan (much is probably missed by just focussing on additive, mostly common, effects).

This is a valid point, and we appreciate the Reviewer not asking for rerunning the three GWAS using a non-additive model. We have added some text to the Discussion section (lines 554-561) to acknowledge this as a limitation of our study.

Reviewer #2:

The final section of the paper aims to assign biological mechanism to the loci identified in the earlier sections. The authors do this by first performing an eQTL colocalisation analysis vs their identified loci, and then running a pathway enrichment scan for the eQTL gene hits. This is a valid hypothesis-generating approach, however I do have a significant concern about the exact procedure used for the pathway enrichment analysis: as it is used currently, I believe that there is a high likelihood of false positive results from this procedure. In a similar vein, due to the solely correlative nature of gene enrichment analysis, the discussion of this result would also benefit from a little moderation.

The hypergeometric test employed by the authors relies on two assumptions which I strongly suspect are violated by these data: equal opportunity for eQTL colocalisation tagging across genes, and independence of gene selection. The authors acknowledge the former (lines 452-455) but do not seem to account for it in the test; for more details of the latter see eg <https://doi.org/10.1093/bioinformatics/btm051> The overall effect of these assumptions being violated is a high likelihood of a false positive result. Given that the results of this analysis lead to a central conclusion of this paper (the association of haem metabolism with ageing), a more careful analysis here is needed, particularly as the haem metabolism finding seems driven almost exclusively by the eQTLs colocalising with a single age-associated locus. The authors should either undertake an analysis that is more robust to these two issues (perhaps employing permutation?), apply an orthogonal approach to confirm the findings of the hypergeometric test, or de-emphasise the pathway association and haem findings, which I see as highly tentative without additional evidence

This is a valid point. However, we note that the P value for the haem metabolism gene set was 1×10^{-24} after adjustment for multiple testing and the cited paper suggests that in the presence of modest correlations (e.g. $r=0.3$) the increase in false positive rates is fairly modest. Nonetheless, we appreciate the Reviewer's suggestion to apply an orthogonal approach to confirm our findings. Therefore, we have performed a Mendelian randomisation (MR) analysis in which we constructed an instrumental variable, using up to 15 independent SNPs for iron-related traits, and tested it against our outcomes. Our findings are consistent with the original hypothesis that haem metabolism may be involved in human ageing. We have added these findings to the manuscript (lines 408-435). Moreover, we have changed our wording (lines 490-493) to make our statement about the involvement of haem metabolism in human ageing less strong (also see our response to the next comment).

Lines 404-410: This wording is too strong given the evidence presented. Even if my technical concern about the hypergeometric test is addressed, gene enrichment analysis provides at

most suggestive evidence of a biological link. A reader might take from this section and the title that haem metabolism is important in human ageing, which is a conclusion that in my opinion is not completely supported by the evidence presented. This section should be moderated to make it clear that although the gene enrichment analysis is suitable as a hypothesis generating tool, it is premature to link haem metabolism with ageing on the basis of these results alone.

We agree with the Reviewer that our wording was too strong, even in the presence of our newly added MR results (see response to previous comment) and have softened the wording in this section (lines 484 and 492) and the Title accordingly.

Line 115: The citation for Therneau, Grambsch, & Fleming (1990) has suffered a typo.

Thank you for noticing this mistake, this has now been fixed.

Line 150: The authors tested gender-specific survival association for lead SNPs by assuming that paternal and maternal effects are independent. This is a surprising assumption to me, and indeed my reading of Timmers 2019 suggests that many loci have sex-concordant effects. It would be helpful if the authors could explain in a little more detail why the assumption of independence was made.

Thank you for pointing this out. We agree with the Reviewer that the effects are usually sex concordant, i.e. the same in fathers and mothers. Our assumption is that the error in measured effects in father's and mother's lifespan is independent. We apologise that the wording was far from clear on this and we have amended it (line 151).

Lines 159-168: It seems a lot of tests were done here -- at least one per locus, plus the combined test of the loci without a significant trend. To aid interpretation of these results the authors should include details of what multiple testing control was applied.

We apologise that we have not explicitly stated the number of tests performed for this analysis and adjusted for this accordingly. Our mention of nominal significant P values in the text could therefore have been misleading. To address this issue, we have now removed the mention of nominal significant associations with age in the main text (although they are still depicted in Figure S8) and instead report the Bonferroni-adjusted P values for *APOE* and the collective test of the remaining loci (lines 343-347). We have now also stated the exact number of tests in the Methods section (lines 166 and 172).

Lines 304-321: Unfortunately confidence interval lines for Figures S7 and S8 were not visible to me, as such I could not fully review this section.

We apologize for this. This has now been fixed.

Lines 386-398: The argument about antagonistic pleiotropy is interesting, but I think that the results of the study are suggestive in this regard, not conclusive. Thus the statement that "it is

clear that loci exhibiting antagonistic pleiotropy exist in humans" is a little strong if based on this study alone, and I suggest a moderation of the claims here. The authors could perhaps address this by discussing with more reference to the recent literature, eg [dx.doi.org/10.1038/s41559-016-0055](https://doi.org/10.1038/s41559-016-0055) and [dx.doi.org/10.1371/journal.pgen.1006328](https://doi.org/10.1371/journal.pgen.1006328).

We agree with the Reviewer that our wording here was too strong and would like to thank him/her for bringing the two recent studies on antagonistic pleiotropy to our attention. We have amended our wording accordingly and added references to the literature suggested by the Reviewer (lines 461-466).

Lines 411-419: In my opinion this paragraph adds little, and strays rather far from the questions posed in the Introduction. Unless a deeper investigation of haem metabolism and ageing is added to the results, I think the paper would be clearer without this paragraph.

Based on the comments of Reviewer #1 and the results from the newly added MR analysis we feel that we have now reasonable evidence that haem metabolism plays a role in healthy ageing. Hence, we decided to keep but adjust this paragraph based on the suggestion of Reviewer #1.

It would be interesting if the authors could link their eQTL findings to the existing body of literature on associations between gene expression and ageing / longevity in humans and model organisms.

Thank you for this suggestion, we have now added a look-up of our cis- and trans-acting genes in a large dataset from Peters et al.¹, which studied the relation of gene expression with age in whole blood (lines 381-393). Moreover, we have added a paragraph to the Discussion in which we link our eQTL findings to ageing research performed in animal models (lines 471-478).

Reviewer #3:

The multivariate analysis of the three GWAS summary statistics resulted an inflation parameter 1.064 from LD regression. After adjusting for this inflation parameter, it is apparently that the two novel loci (SLC4A7 and LINCO2513) are no longer genome wide significant. In addition, the P values of the novel loci are all close to 5E-8. Additional replication analysis is necessary to verify these novel loci.

We appreciate the point, with some reservations. Firstly, where the LD score regression intercept is modest, it is not uncommon to make no adjustment of the form suggested, including publications in this journal^{2,3}. Secondly, the authors have spent the last five years trying to gather cohorts to perform these sorts of analyses and further data at the scale needed is simply not available at present. For example, to attain a P value $< 1 \times 10^{-3}$ for the two loci would require in excess of 300,000 parental lives at replication, if the estimated effect size is correct and replicated. Nonetheless, we accept the underlying fundamental point that results that are close to the genome-wide significant threshold should not be accepted unthinkingly and have thus adjusted the wording of the Results to reflect this (lines 285-287, 318-320). We

would also like to note that both the *SLC4A7* and *LINC02513* locus have previously been implicated in age-related diseases, which we already had mentioned in our manuscript. Moreover, based on a comment of Reviewer #2, we now show that the *LINC02513* locus may serve as a master regulator of age-related transcription in whole blood. This indicates that there is additional evidence for a role of both loci in healthy ageing.

The manuscript concluded that there is antagonistic pleiotropy among the loci identified. However, this is just a speculation. There is no clear evidence from the current analysis. Although combining multiple traits will improve statistical power for detecting associated loci, the MANOVA analysis cannot differentiate pleiotropy or colocalization. The evidence is quite weak to claim that a genetic variants are associated with all three traits by restricting the variants with a P value < 0.05 for all three traits.

Here there has been a misunderstanding as to what we meant, for which we apologise. We were not suggesting antagonistic pleiotropy amongst the three analysed traits (i.e. parental lifespan, healthspan and longevity - “healthy life”), but rather about antagonistic pleiotropy between healthy life variants and other traits (coming from other GWAS). However, the point regarding distinguishing between pleiotropy and colocalization is still relevant and we have therefore amended our wording of the paragraph in the Discussion (lines 461-463), also taking into account the comments from Reviewer #2. Moreover, for consistency, we have removed the part discussing the ZW10 locus, given that this was based on non-genome-wide significant results from other GWAS.

In figure 1, how the tree in c) was obtained? The sample size of the Longevity GWAS was the smallest. This can affect the estimation of genetic correlations.

We indeed forgot to mention how the tree in Figure 1 was obtained. Hence, we have now clarified this in the figure legend of Figure 1. Moreover, we have added a note to this legend mentioning that the number of significant genetic correlations with longevity is lower due to the smaller sample size of this GWAS.

Page 4, line 113 and 114, $\beta_{1,2,\dots,n}$ should be $\beta_1, \beta_2, \dots, \beta_n$ and $X_{1,2,\dots,n}$ should be X_1, X_2, \dots, X_n .

Thank you for noticing this mistake, this has now been fixed.

References

1. Peters, M. J. *et al.* The transcriptional landscape of age in human peripheral blood. *Nat. Commun.* **6**, 8570 (2015).
2. Davies, G. *et al.* Study of 300,486 individuals identifies 148 independent genetic loci influencing general cognitive function. *Nat. Commun.* **9**, 1–16 (2018).
3. Visconti, A. *et al.* Genome-wide association study in 176,678 Europeans reveals genetic loci for tanning response to sun exposure. *Nat. Commun.* **9**, 1–7 (2018).

Reviewer #1 (Remarks to the Author):

Thank you for your thoughtful responses, and improvements to the manuscript. I have no further comments or concerns, other than that the authors should include the results of a "leave one out" analysis in the new iron-related MR section to show the results are independent of HFE haemochromatosis (i.e. it is natural variation in iron that is associated - it is not being driven by disease-causing variants having an overwhelming effect).

Dr Luke C. Pilling

Reviewer #2 (Remarks to the Author):

The authors have addressed all my concerns.

Reviewer #3 (Remarks to the Author):

The authors generally addressed my concerns well. I have some additional comments.

- 1) Page 5, line 151, σ_{12} and σ_{22} need to be defined.
- 2) Page 5, line 159, the formula for SE_{β} is not correct. The denominator of the second term inside the square root should be β_{2all}
- 3) I have some difficulty to understand the MR results. The authors mentioned that there was evidence of violation of the pleiotropy assumption (page 11). However, in Figure S11, the intercepts of MR-Egger seem not significantly depart from 0. How the pleiotropy assumption was violated? Have the authors performed the outlier test, for example using MR-PRESSO? In Table S10, the causal effect estimates were very small (ranged from -0.03 to 0.01), suggesting no causal relationship. However, the causal effect estimates were large in Table 3 and ranged from -0.01 to 0.80. How to explain the results? In general, it has been suggested that interpretation of MR results should be cautious. (Burgess S, Smith GD, Davies NM, Dudbridge F, Gill D, Glymour MM, P. Hartwig FP, Holmes MV, Minelli C, L. Relton CL, Theodoratou E: Guidelines for performing Mendelian randomization investigations. Wellcome Open Research 2020.)

We thank the Reviewers for their acceptance of our revisions and hope the following addresses the outstanding issues.

Reviewer #1

Thank you for your thoughtful responses, and improvements to the manuscript. I have no further comments or concerns, other than that the authors should include the results of a "leave one out" analysis in the new iron-related MR section to show the results are independent of HFE haemochromatosis (i.e. it is natural variation in iron that is associated - it is not being driven by disease-causing variants having an overwhelming effect).

We thank the Reviewer for contacting us regarding this point and have added the results of the requested analysis to the relevant sections (lines 237-238 and 429-430).

Reviewer #3:

Page 5, line 151, σ_{12} and σ_{22} need to be defined.

Thank you for spotting this. We have replaced σ_{12} and σ_{22} with SE to match the following equation and defined them in the text (line 151).

Page 5, line 159, the formula for $SE\alpha$ is not correct. The denominator of the second term inside the square root should be β_{2all}

Thank you for noticing this mistake, this has now been fixed.

I have some difficulty to understand the MR results. The authors mentioned that there was evidence of violation of the pleiotropy assumption (page 11). However, in Figure S11, the intercepts of MR-Egger seem not significantly depart from 0. How the pleiotropy assumption was violated?

We apologise for not making this clear. The pleiotropy assumption was only violated for transferrin saturation, not the other traits. We have made this more explicit in the Results section (lines 422-426).

Have the authors performed the outlier test, for example using MR-PRESSO?

We have now included a leave-one-out analysis, consistent with the suggestion from Reviewer 1 and the Burgess et al. paper mentioned by the Reviewer. Further discussion of Burgess et al's advice has been included in the Discussion section (lines 472-483).

In Table S10, the causal effect estimates were very small (ranged from -0.03 to 0.01), suggesting no causal relationship. However, the causal effect estimates were large in Table 3 and ranged from -0.01 to 0.80. How to explain the results?

We believe this is a consequence of homeostasis and outline the argument in the Results section (lines 435-438).

In general, it has been suggested that interpretation of MR results should be caution. (Burgess S, Smith GD, Davies NM, Dudbridge F, Gill D, Glymour MM, P. Hartwig FP, Holmes MV, Minelli C, L. Relton CL, Theodoratou E: Guidelines for performing Mendelian randomization investigations. Wellcome Open Research 2020.)

We agree and have acknowledged the salient points in our Discussion section (lines 472-483). Moreover, we confirm that other more prosaic issues (e.g. variant harmonisation) have been dealt with properly.